# An exploratory survey about using ChatGPT in education, healthcare, and research

**Mohammad Hosseini**[1][☉]*, **Catherine A. Gao**[2☉], **David M. Liebovitz**[3,4], **Alexandre M. Carvalho**[5,6], **Faraz S. Ahmad**[1,7,8], **Yuan Luo**[1], **Ngan MacDonald**[9], **Kristi L. Holmes**[1,9,10], **Abel Kho**[3,9]

1 Department of Preventive Medicine, Northwestern University Feinberg School of Medicine, Chicago, Illinois, United States of America, 2 Division of Pulmonary and Critical Care, Department of Medicine, Northwestern University Feinberg School of Medicine, Chicago, Illinois, United States of America, 3 Divisions of General Internal Medicine and Health and Biomedical Informatics, Department of Medicine, Northwestern University Feinberg School of Medicine, Chicago, Illinois, United States of America, 4 Center for Medical Education in Digital Health and Data Science, Northwestern University Feinberg School of Medicine, Chicago, Illinois, United States of America, 5 Division of Infectious Diseases, Department of Medicine, Northwestern University Feinberg School of Medicine, Chicago, Illinois, United States of America, 6 Center for Pathogen Genomics & Microbial Evolution, Northwestern University Feinberg School of Medicine, Chicago, Illinois, United States of America, 7 Division of Cardiology, Department of Medicine, Northwestern University Feinberg School of Medicine, Chicago, Illinois, United States of America, 8 Bluhm Cardiovascular Center for Artificial Intelligence, Northwestern Medicine, Northwestern University Feinberg School of Medicine, Chicago, Illinois, United States of America, 9 Institute for Artificial Intelligence in Medicine, Northwestern University Feinberg School of Medicine, Chicago, Illinois, United States of America, 10 Galter Health Sciences Library and Learning Center, Northwestern University Feinberg School of Medicine, Chicago, Illinois, United States of America

☉ These authors contributed equally to this work.
* mohammad.hosseini@northwestern.edu

**Data Availability Statement:** Data are at https://zenodo.org/record/7789186#.ZCb0eezML0o Code are available at https://github.com/cloverbunny/gptsurvey/blob/main/gptsurvey.ipynb.

## Abstract

### Objective

ChatGPT is the first large language model (LLM) to reach a large, mainstream audience. Its rapid adoption and exploration by the population at large has sparked a wide range of discussions regarding its acceptable and optimal integration in different areas. In a hybrid (virtual and in-person) panel discussion event, we examined various perspectives regarding the use of ChatGPT in education, research, and healthcare.

### Materials and methods

We surveyed in-person and online attendees using an audience interaction platform (Slido). We quantitatively analyzed received responses on questions about the use of ChatGPT in various contexts. We compared pairwise categorical groups with a Fisher's Exact. Furthermore, we used qualitative methods to analyze and code discussions.

### Results

We received 420 responses from an estimated 844 participants (response rate 49.7%). Only 40% of the audience had tried ChatGPT. More trainees had tried ChatGPT compared with faculty. Those who had used ChatGPT were more interested in using it in a wider range of contexts going forwards. Of the three discussed contexts, the greatest uncertainty was

**Funding:** This work was supported in part by the Northwestern University Institute for Artificial Intelligence in Medicine. CAG is supported by NIH/NHLBI F32HL162377. KH and MH are supported by the National Center for Advancing Translational Sciences (NCATS, UL1TR001422), National Institutes of Health (NIH). FSA is supported by grants from the National Institutes of Health/National Heart, Lung, and Blood Institute (K23HL155970) and the American Heart Association (AHA number 856917). The funders have not played a role in the design, analysis, decision to publish, or preparation of the manuscript.

**Competing interests:** The authors report no conflicting interests.

shown about using ChatGPT in education. Pros and cons were raised during discussion for the use of this technology in education, research, and healthcare.

## Discussion

There was a range of perspectives around the uses of ChatGPT in education, research, and healthcare, with still much uncertainty around its acceptability and optimal uses. There were different perspectives from respondents of different roles (trainee vs faculty vs staff). More discussion is needed to explore perceptions around the use of LLMs such as ChatGPT in vital sectors such as education, healthcare and research. Given involved risks and unforeseen challenges, taking a thoughtful and measured approach in adoption would reduce the likelihood of harm.

## Introduction

The introduction of OpenAI's ChatGPT has delivered large language model (LLM) systems to a mainstream audience. Other applications such as Elicit, SciNote, Writefull, and Galactica, have previously existed, but the exponential growth of ChatGPT's audience has sparked vigorous discussions in academic circles. LLMs have demonstrated remarkable ability (and sometimes inability) in generating text in response to prompts. Some LLMs like Elicit and Med-PaLM can scan available literature and suggest specific questions or insights about a particular topic/question by leveraging available knowledge. The new GPT4 can also learn from images, thereby multiplying possible use cases of LLM, especially in education, healthcare and research settings where visual representations are fundamental to create or enhance understanding. To explore the implications of using LLMs in research, education and healthcare, Northwestern University's Institute for Artificial Intelligence in Medicine (I.AIM) and Institute for Public Health & Medicine (IPHAM) organized a hybrid (virtual and in-person) event on Feb 16[th] 2023 entitled "Let's ChatGPT". This event consisted of lively discussions and an exploratory survey of participants. In this article, we present survey results and provide a qualitative analysis of raised issues.

### Using ChatGPT and other LLMs in education

Responses to the use of ChatGPT in education are varied. For instance, some New York schools banned students from using ChatGPT [1], while others adopted policies in their syllabus that encourage students to engage with these models as long as they disclose it [2]. Some educators fed ChatGPT questions from a freely available United States Medical Licensing Examination (USMLE) and reported a near or at passing range performance [3]. Others have suggested that using ChatGPT facilitates personalized and interactive learning, can improve assessment and create an ongoing feedback loop to inform teaching and learning [4–6]. It can also create opportunities in specific contexts. For example, in law where original references might be complicated to comprehend for students, ChatGPT could help in reciting complicated laws and making them more understandable [7]. In teaching computer science in Harvard's flagship CS50 course, systems similar to ChatGPT will help students explain and debug their code, improve design, and answer questions, making it more likely "to approximate a 1:1 teacher:student ratio for every student" [8]. Delegating these tasks to AI is believed to free up

teaching fellows' time to engage in more meaningful and interpersonal interactions with students and focus on providing qualitative feedback [8].

As the technology improves, the debate is still open about ethical and educational uses, with many issues remaining unresolved and concerns being explored. Among such concerns, the issue of "disguising biases" is noteworthy. It is believed that by weaving information found in various sources (some of which could be biased), ChatGPT creates a "tapestry of biases", thereby making it more difficult to pinpoint the origins of any specific bias in the educational resources it produces [9]. Using ChatGPT increases the likelihood of plagiarism, can lead to the inclusion of irrelevant or inaccurate information in students' essays, and presents challenges in assessing students' work [10]. The use of ChatGPT in education has been reviewed in more detail by others [11, 12].

## Using ChatGPT and other LLMs in healthcare

There has long been excitement around the use of Artificial Intelligence (AI) in healthcare applications [13]. Applications of interest for language-specific tools include improving the efficiency of clinical documentation, decreasing administrative task burdens, clarifying complicated test result reports for patients, and responding to in-basket Electronic Medical Record (EMR) messages. For example, Doximity has released a beta version of DocsGPT, a tool that integrates ChatGPT to assist clinicians with tasks such as writing insurance denial appeals [14]. There has also been reports about using ChatGPT to answer medical questions [15], write clinical case vignettes [16], and simplify radiology reports to enhance patient-provider communication [17]. The electronic health record system, Epic, has announced they are examining pilot programs to use this technology for drafting notes and replying to in-basket messages [18].

In deliberations about using LLMs in healthcare, a major caveat lies in the models' tendency to 'hallucinate' or 'confabulate' factual information, which given the sensitivity of this context, could be extremely dangerous. Accordingly, the importance of having the output reviewed by domain experts (e.g., for accuracy, relevance, and reliability) cannot be overemphasized. Furthermore, before using LLMs in healthcare it is crucial to understand their biases. Depending on the quality of the training data and employed reinforcement feedback processes, different LLMs might have dissimilar biases that users should be aware of. The use of ChatGPT and other LLMs has been reviewed by others in more detail [19–21].

## Using ChatGPT and other LLMs in research

Even before the introduction of OpenAI's ChatGPT, computer generated text was used in academic publications. As of 2021, the estimated prevalence was 4.29 papers for every one million papers [22], raising concerns about the negative impact of using LLMs on the integrity of academic publications [23]. One way the community was able to detect these papers was through spotting so-called tortured phrases (i.e., the AI-generated version of an established phrase used in specific disciplines for certain concepts and phenomena).

ChatGPT, on the other hand, generates fluent and convincing abstracts that are difficult for human reviewers or traditional plagiarism detectors to identify [24]. As ChatGPT and other recently developed applications based on LLMs mainstream the use of AI-generated content, detection will likely become much more difficult. This is partly because, (1) with an increase in the number of users, LLMs learn quicker and produce better human-like content, (2) more recent LLMs benefit from better algorithms and, (3) researchers are more aware of LLMs' shortcomings e.g., use of tortured phrases and will likely mix generated content with their own writing to disguise their use of LLMs. Detection applications seem unreliable and for the

foreseeable future will likely remain so. Given challenges of detecting AI-generated text, it makes sense to err on the side of transparency and encourage disclosure. Various journal editors and professional societies have developed disclosure guidelines, stressing that LLMs cannot be authors [25, 26], and, when used, should be disclosed in the introduction or methods section, describing who used the system, when, and using which prompts, as well as among cited references [27, 28].

Besides assisting researchers in improving their writing [29], LLMs can also be used in scholarly reviews to support editorial practices. For example, by supporting the search for suitable reviewers, the initial screening of manuscripts, and the write-up of final decision letters from individual review reports. However, various risks such as inaccuracies and biases as well as confidentiality concerns require researchers and editors to engage with LLMs cautiously [30]. The use of ChatGPT and other LLMs in research has been reviewed in detail by others [21, 31, 32].

## Methods

The research protocol and the first draft of survey questions were developed (M.H. and C.A.G) based on available and ongoing work about LLMs and ChatGPT, with suggestions from other panel members (K.H. and N.K.N.M) and a team member (E.W.). ChatGPT was used to brainstorm survey questions. D.L. used OpenAI ChatGPT on the 27th of January 2023 at 6:06pm CST using the following prompt: "please create survey questions for medical students, medical residents, and medical faculty members to answer regarding ideas for use and attitudes surrounding use of ChatGPT in education and research" [33]. The Northwestern IRB granted an exemption (STU00218786). We received permission from the Vice Dean of Education to gather responses from medical trainees attending the session. Attendees were informed about the survey details, such as anonymized data collection and voluntary participation, and were offered a chance to view the information sheet and consent form before the start of the survey. We included a slide with the summary details as well as a QR code linking to additional details about the proposed study. Our IRB protocol requested a waiver of participants' verbal or written consent because this was a hybrid event with online and in-person attendees, which made both forms of consent impractical. Upon IRB's approval of this waiver, attendees were informed that by logging in to the Slido polling platform, they were consenting to participate in our study. We collected anonymized and unidentifiable data using a paid version of Slido (Bratislava, Slovakia; https://www.sli.do/). The full survey is available in the S1 File.

The quantitative survey data were analyzed and visualized (C.A.G) in python v 3.8 with scipy v1.7.3, matplotlib v3.5.1, seaborn v0.11.2, tableone v0.7.10 [34], and plot_likert v0.4.0 [35]. ChatGPT was used for minor code troubleshooting. For the small subset of 18 respondents who selected multiple roles, we took their most senior role and most clinical role for analysis. Binarized responses included any answer with 'yes', with the other category being 'No + unsure'. Categories were compared pairwise using Fisher's Exact tests.

The discussion was analyzed after transcribing the session (M.H.). For this purpose, we used the three topic areas highlighted in the event description (education, healthcare and research) to qualitatively code the transcripts using an inductive approach [36]. Using these codes we analyzed the transcript. Subsequently, we identified three subcodes within each code (possible positive impacts, possible negative impacts and remaining questions), bringing the total number of codes to nine. Using these nine codes, we analyzed the transcript for a second time and generated a report. Upon the completion of the first draft of the report, feedback was sought from all members of the panel and the text was revised accordingly.

## Results

### Survey results

We had 1,174 people register for the event. The peak number of webinar participants during the event was 718, and 126 people indicated they would attend in-person. We received survey responses from 420 people; a conservative estimated response rate is 49.7%. The smallest group was medical trainees (medical students, residents, and fellows) at 14 respondents (3.3% of all respondents), and the second smallest was clinical faculty with 45 (10.7%) respondents (Table 1). There were more research trainees (graduate students and postdoctoral researchers) with 53 (12.6%) respondents, and research faculty with 65 (15.5% respondents). Administrative staff made up 70 (16.7%) of respondents. The largest group of respondents identified as 'Other', with 173 respondents (41.2% of all respondents).

Overall, only 40% of the audience had tried ChatGPT. Medical and research trainees were more likely to have used ChatGPT compared with faculty and staff. Significantly more medical trainees (medical student, residents, fellows) had tried ChatGPT (64.2%) compared with clinical faculty (31.1%), p = 0.03. The percentage of graduate students and postdoctoral researchers who had tried ChatGPT (56.6%) was closed to that of research faculty (49.2%), p = 0.46.

Across all roles except medical trainees, the most common response regarding interest in using ChatGPT going forwards was 'Somewhat'. Those who had used ChatGPT already had higher interest in using it compared with those who had not; 39.9% had interest in using it 'to a great extent' compared to 15.9% who had no interest in using it (p<0.001; Fig 1).

In response to questions about whether ChatGPT can be used in specific contexts, there was greater uncertainty around its use in Healthcare and Education, compared to using it in Research (Table 2). For Research, only 75 (17.9%) of respondents selected 'I don't know, it is too early to make a statement', compared with 226 (53.8%) when asked about using it in Education (p<0.001), and 177 (42.2%), when asked about using it in Healthcare (p<0.001). Medical and research trainees were more interested in using it for education purposes compared with clinical and research faculty, though this was not statistically significant. Of note, when responding to the question about using ChatGPT in Healthcare, a significant portion (42% of respondents) of respondents approved of using it for administrative purposes (for example, writing letters to insurance companies) and there was a smaller population of respondents who thought it could be used for any purpose (12.2%). More medical trainees felt it was acceptable to use this technology for healthcare purposes (including administrative purposes), compared with clinical faculty 92.9% 'yes' vs 48.9% 'yes', p = 0.004 (Fig 2).

Those who had already used ChatGPT were more likely to deem it acceptable for research purposes (89.3% 'yes') versus those who had not used it before (75% 'yes'), 14.3% higher, p<0.001 (Fig 3). Similarly, those with prior experience thought it was acceptable to use in

**Table 1. Number of respondents, by role, and whether they had used LLMs before.**

| Respondent Role | Number | Used LLM before, n (%) | |
|---|---|---|---|
| | | No | Yes |
| Medical Student, Resident, Fellow | 14 | 5 (35.7) | 9 (64.3) |
| Graduate Student, Postdoc Researcher | 53 | 23 (43.4) | 30 (56.6) |
| Clinical Faculty | 45 | 31 (68.9) | 14 (31.1) |
| Research Faculty | 65 | 33 (50.8) | 32 (49.2) |
| Administrative Staff | 70 | 48 (68.6) | 22 (31.4) |
| Other | 173 | 112 (64.7) | 61 (35.3) |
| Total | 420 | 252 (60.0) | 168 (40.0) |

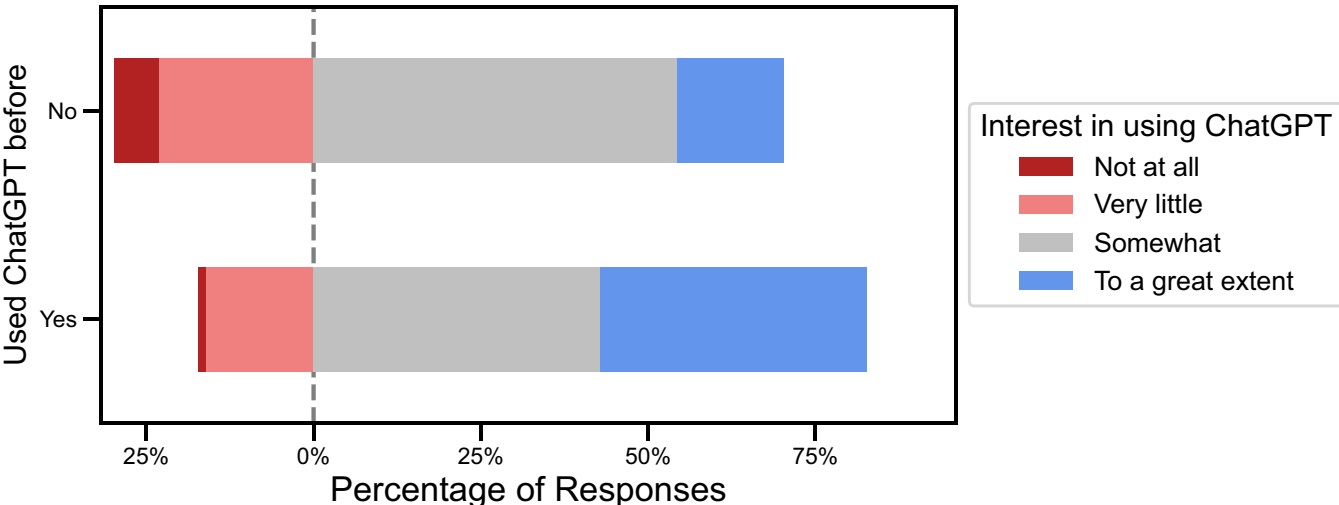

**Fig 1. Interest in using ChatGPT as broken down by previous usage.** There was greater interest going forwards among those who had already tried ChatGPT compared to those who had not.

healthcare 62.5% vs 48.8%, 13.7% higher, p = 0.008. They also thought it was more acceptable to use in education, 63.9% vs 30.2%, 33.7% higher, p<0.001.

### Analysis of the Q&A session

**Education.** *Possible positive impacts.* "Leveling the playing field" for students with different language skills was identified as an advantage of using LLMs. Since students' scientific abilities should not be overshadowed by their insufficient language skills, ChatGPT was seen as a solution that could help fix errors in writing and accordingly, an instrument that can support students who might be challenged by writing proficiency—specifically those not writing in their native language. Another useful application was "adding the fluff" to writing (i.e., details that could potentially improve comprehension), especially for those with communication challenges. Structuring and summarizing existing text or creating the first draft of letters of application with specific requirements were also mentioned among possible areas where ChatGPT could help students. Another mentioned possibility was to use ChatGPT as a studying tool that

**Table 2. Survey response breakdown, by topic of education, research, and healthcare.**

| Topic | Statement | Response, n (%) |
| --- | --- | --- |
| Education | I don't know, it is too early to make a statement | 226 (53.8) |
| | No, it should be banned | 11 (2.6) |
| | Yes, it should be actively incorporated | 183 (43.6) |
| Research | I don't know, it is too early to make a statement | 75 (17.9) |
| | No, it should not be used at all | 6 (1.4) |
| | Yes, as long as its use is transparently disclosed | 259 (62.0) |
| | Yes, but it should only be used to help brainstorm | 68 (16.3) |
| | Yes, disclosure is NOT needed | 10 (2.4) |
| Healthcare | I don't know, it is too early to make a statement | 177 (42.2) |
| | No, it should not be used at all | 15 (3.6) |
| | Yes, it can be used for administrative purposes | 176 (42.0) |
| | Yes, it can be used for any purpose | 51 (12.2) |

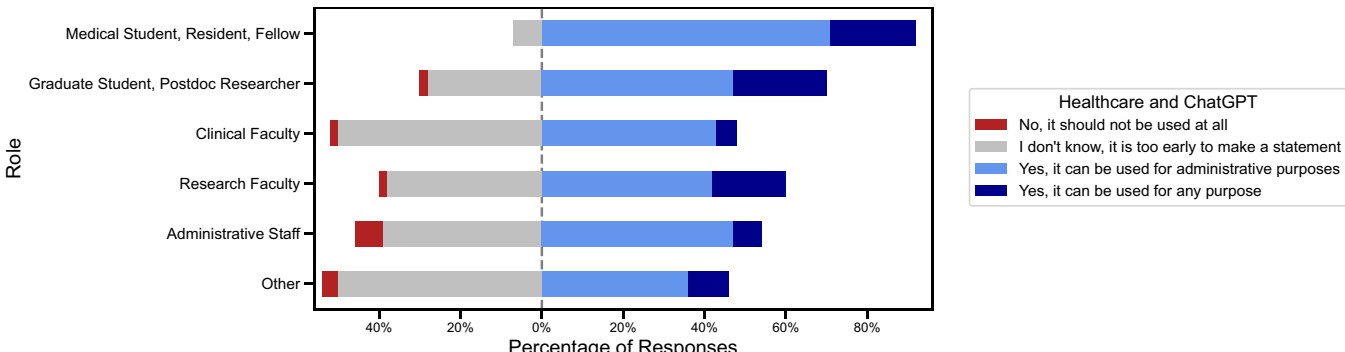

**Fig 2. Use of ChatGPT in healthcare, by respondent role.** Breakdown of proportions of answers when asking about ChatGPT use in Healthcare, as split by respondent's role. Students had higher acceptability of ChatGPT's use than faculty and staff.

(upon further improvements and approved accuracy) could describe medical concepts at a specific comprehension level (e.g., "explain tetralogy of fallot at the level of a tenth grader").

*Possible negative impacts.* Given existing inaccuracies in content generated by systems such as ChatGPT, a panel member warned medical students against using them to explain medical concepts and encouraged them to have everything "double and triple checked". To the extent that ChatGPT could be used to find fast solutions, and as a substitute for hard work and understanding the material (e.g., only to get through the assignments or take shortcuts), it was believed to be harmful for education. Clinical-reasoning skills were believed to be at risk if ChatGPT-like systems are used more widely. For instance, it was believed that writing clinical notes helps students "internalize the clinical reasoning that goes into decision making", and so until such knowledge is cemented, using these systems would be harmful for junior medical students. One member of the audience warned that since effective and responsible use of ChatGPT requires adjusted curricula and assessment methods, employing them before these changes are enacted would be harmful. A panel member highlighted the lack of empirical evidence in relation to the usefulness and effectiveness of these systems when teaching different cohorts of students with various abilities and interests. As such, early adoption of these systems in all educational contexts was believed to have unforeseen consequences.

*Remaining questions.* Challenges of ensuring academic integrity and students' willingness to disclose the use of ChatGPT were raised by some attendees. However, as a clinical faculty member suggested, these are neither new challenges nor unique problems associated with

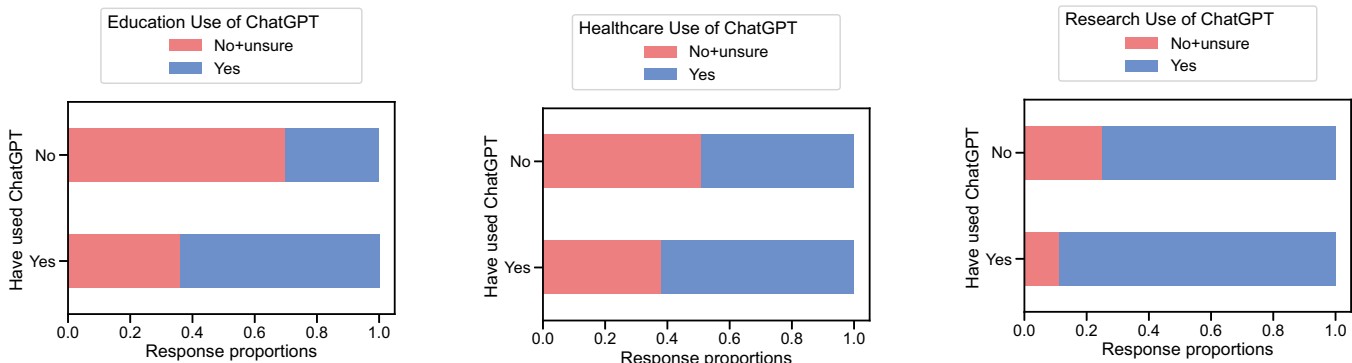

**Fig 3. Acceptability of ChatGPT use.** Comparison of binarized responses in use of ChatGPT for education, healthcare, and research, broken down by previous use of ChatGPT.

ChatGPT because even in the absence of such tools, one could hire somebody to write essays. Plagiarism detection applications and stricter regulations have not deterred outsourcing essay writing. Therefore, it remains an open question as to how ChatGPT changes this milieu.

A panelist suggested that similar to when ChatGPT is used to write code (e.g., in Python) and the natural tendency to test generated code to see if it actually works (e.g., as part of the larger code), students should employ methods to test and verify the accuracy and veracity of generated text. However, since systems like ChatGPT are constantly evolving, developing suggestions and guidelines for verification is challenging.

Information literacy was another issue raised by a panelist. New technologies such as ChatGPT extend and complicate existing discussions in terms of how information is accessed, processed, evaluated and ultimately consumed by users. From a university library perspective, training and supporting various community members to responsibly incorporate new technology in decision making and problem solving requires mobilizing existing and new resources.

**Healthcare.** *Possible positive impacts.* Improving communication between clinicians and patients was among anticipated possible gains. For example, it was highlighted that "doctors might not be in their best self" during an extremely busy week when they are responding to patient's EMR messages, and so ChatGPT could ensure that all niceties are there, include additional content based on patients' history and maintain emotional consistency in communication. Upon further development, these systems could help centralize and organize patient records by flagging areas of concern to improve diagnosis and effective decision making. Currently, our medical records lack sufficient usability and when assessing patients, one is concerned that some vital information might be "buried in a chart" that is not readily accessible. However, with LLMs acting as "assistants" or "co-pilots", able to find these hidden and sometimes critical pieces of information, it could be possible for the provider to save time and improve care delivery.

Efficiency of documentation was highlighted as a potential gain for clinicians, patients and the healthcare system. For example, increased efficiency in note-taking through prepopulation of forms, voice recording and morphing that into clinician notes, and synthesizing existing patient notes to save clinicians' time were noted as possibilities. This increased efficiency was believed to benefit patients through improved care and increased patient-clinicians interaction time, which could improve shared decision-making conversations. One panelist highlighted that patient notes are logged in the EHR system mostly late at night or outside regular working hours, stressing the burden of note taking on clinicians as a driver of burnouts.

*Possible negative impacts.* Given available evidence about ChatGPT's inaccuracies and so-called hallucinated content [37] as well as lack of transparency about the used sources in training it, using these systems in triage and admission of new patients or for clinical diagnosis was deemed risky. One panelist highlighted previous failures of AI models in clinical settings [38, 39] as a lesson for the community to adopt these technologies with caution and only after regulatory approvals. Furthermore, the COVID-19 pandemic and clinicians' experience of having to fight "malicious misinformation" was used as an example to highlight risks associated with irresponsible use. Malevolently using wrong or inaccurate data to train an LLM was described as "poisoning the dataset" to produce a predictive model that generates erroneous information.

Although many viewed the speculated positive impact on efficiency favorably, some shared reservations about it, highlighting that the freed-up time could be seen as an opportunity to ask clinicians to visit more patients instead of spending more time with them. The explanation was that the healthcare system could redirect an opportunity like this to generate additional revenue. Furthermore, using technology to consolidate existing notes or pre-populate forms was believed to increase the likelihood that falsehood could be copy-pasted and result in

carrying forward errors. The concern being that since these systems have the propensity to pass on information *as well as misinformation*, wrong diagnoses could be carried forward without being questioned. Unless the veracity of carried historical information is questioned, clinicians might be trained out of the habit of critical thinking and assume all information as reliable.

*Remaining questions.* When discussing incorporation of ChatGPT in healthcare, specific techno-ethical challenges were highlighted. For example, it was stressed that while excitement about technology is positive, specific aspects need profound deliberation and intentional design. These include defining and enforcing different access levels (e.g., to clinical notes), regulating data reuse, protecting patients' privacy, accountability of user groups, and credit attribution for data contributions. Furthermore, securing the required financial investment to responsibly incorporate LLMs into existing information technology infrastructure and workflows was believed to be challenging.

Upon debating as to whether ChatGPT is a friend or foe, one panelist mentioned challenges such as distribution disparities, and said "unfortunately, the track record of our use of technologies is not strong. New technologies have always worsened disparities and I have a significant concern that the computer power that is needed to generate and power these systems will be inadequately distributed".

When discussing the risk of malevolently poisoning LLMs' training data, one panelist highlighted that it remains unclear how healthcare data should be curated for LLMs and how erroneous information could be identified and removed. Furthermore, who should be responsible to monitor the sanctity of training data or prioritize available information (e.g., based on the reliability of used sources)? It was noted that when using tools such as the Google search engine, users have already developed specific skills to question unique sources but because ChatGPT "assimilates" enormous amounts of information, attributions are ambiguous and so verification remains challenging.

**Research.** *Possible positive impacts.* Refining scholarly text or making suggestions to improve existing texts were highlighted among possible positive impacts. Support provided by a writing center were used as an analogy to describe some of these gains. One unique feature of ChatGPT was believed to be bidirectional communication, which allows (expert) users to "interrogate the system and help refine the output", which will ultimately benefit all users in the long run.

*Possible negative impacts.* Lack of transparency about the used data to train LLMs was believed to hide biases and disempower researchers in terms of "grasping the oppression that has gone into the answers". This issue was also stressed by a member of the audience who questioned the language of used sources. One panelist speculated that the training data likely contained more sources in overrepresented languages within the scholarly corpus (e.g., English, French). Furthermore, since ChatGPT is currently made unavailable (by OpenAI) in countries such as China, Russia, Ukraine, Iran and Venezuela, it cannot be trained by or receive feedback from researchers who are based in these countries, and thus, might be biased towards the views of researchers based in specific locations.

*Remaining questions.* One member of the audience believed that disclosure guidelines (e.g., have researchers disclose what part of the text is influenced by ChatGPT) are unenforceable and so, their promotion is moot. They added that the existing norms on plagiarism cover potential misconduct using ChatGPT. One panelist agreed with the unenforceability of guidelines (because researchers may alter AI-generated text to disguise their use), but highlighted that given the novelty of ChatGPT and its unique challenges, good practices in relation to this technology should be specified and promoted nonetheless.

We asked attendees to describe the most important risks and benefits of using ChatGPT with only one keyword. After correcting typographical errors and replacing all plurals with singular words (with the help from ChatGPT), we used a free online word cloud generator [40] to produce the following two figures (Fig 4A and 4B).

## Discussion

We hosted a large forum to gauge interest and explore perspectives about using ChatGPT in education, research, and healthcare. Overall, there was significant uncertainty around the acceptability of its use, with a large portion of respondents saying it was too early to make a statement and that they remained somewhat interested in using ChatGPT.

As demonstrated by their positive opinions and favorable thoughts, trainees were more interested in using ChatGPT than faculty. Moreover, more trainees than faculty had already tried ChatGPT. This points to a potential generational divide between early adopters (trainees) and late adopters (faculty), with the latter in positions of power to dictate policy to trainees and the academic community at large. As trainees are likely to be more actively engaged with this technology over the years, it is important for senior faculty to also experiment with the technology. Critically, shared decision-making about appropriate use must incorporate the voices of all user groups, with emphasis on the input of trainees, which are the ones most likely to be impacted by the continued development and deployment of this nascent technology.

Our results showed that in terms of regulating the use of LLMs, a one-size-fits-all strategy might not work. For example, more respondents indicated that it is acceptable to use LLMs for research and healthcare (including for administrative tasks) than for educational purposes. Context specific policies may be helpful in clarifying what is deemed acceptable use, so as to avoid miscommunication or ambiguity. Future studies could examine each use case in greater detail, specifically among the population of potential users.

Given LLMs' potential to be integrated in different context, it is reasonable to encourage different cohorts to explore and test this technology. Only 40% of our respondents had tried ChatGPT. It is important to note that participants who had used this technology had a more optimistic outlook about LLMs in general whereas never-users seemed to have more concerns about its widespread adoption. Thus, it is important to continue to educate and inform different cohorts about LLMs and their responsible use through practical applications (including live demonstrations), so never-users can grasp the technology and help dispel the fear of the unknown. Using and engaging with LLMs is essential to learning about their abilities and limitations.

Respondents and audience members had a wide range of interesting points with regards to the use of ChatGPT for research, education, and healthcare, with a mixture of positive and negative responses. Ongoing discussion is essential, especially given the current "black-box" nature of ChatGPT and other LLMs, with users left in the dark on *how* outputs are generated. Unresolved questions remain about how LLMs curate content, the corpus of data they are trained on, the weights used to sort and prioritize evidence, and the risks of spreading fake news, misinformation or bias. One potential solution from legislators would be to require increased transparency from OpenAI and other LLM companies.

### Limitations

Some of the limitations of this study include our inability to break down and better delineate the large "Other" category of respondents. Since respondents were likely interested in ChatGPT to register for and attend the event, and also complete the survey, our results might not be representative of the various cohorts within the academic community.

A

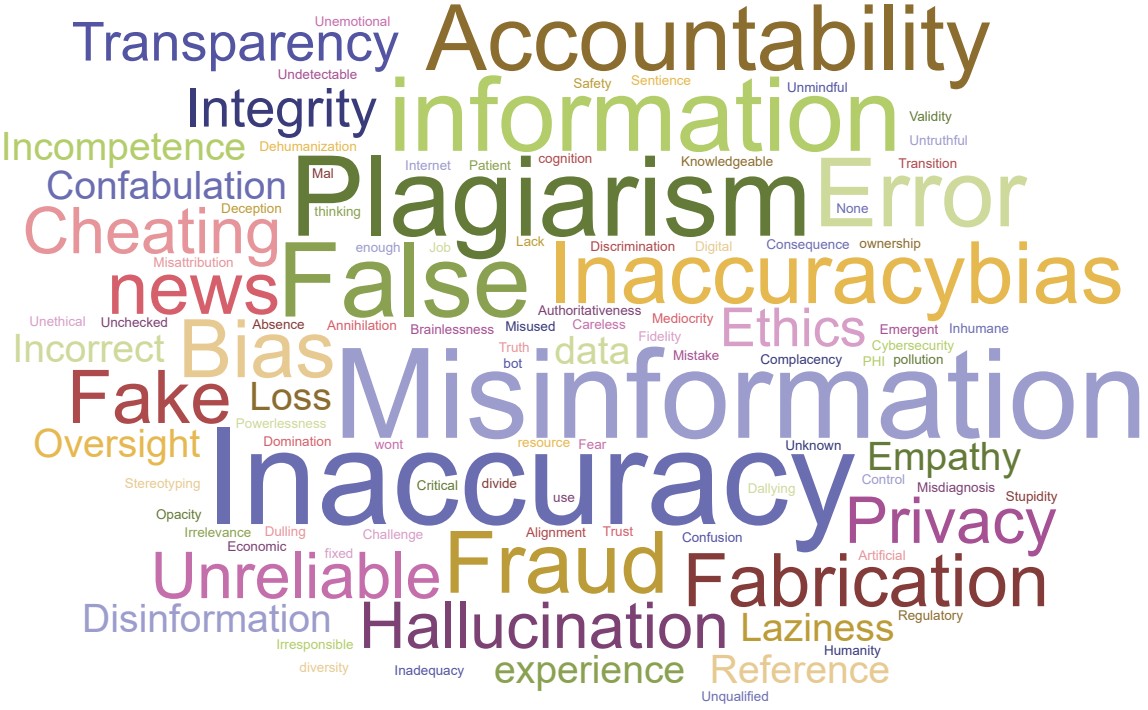

B

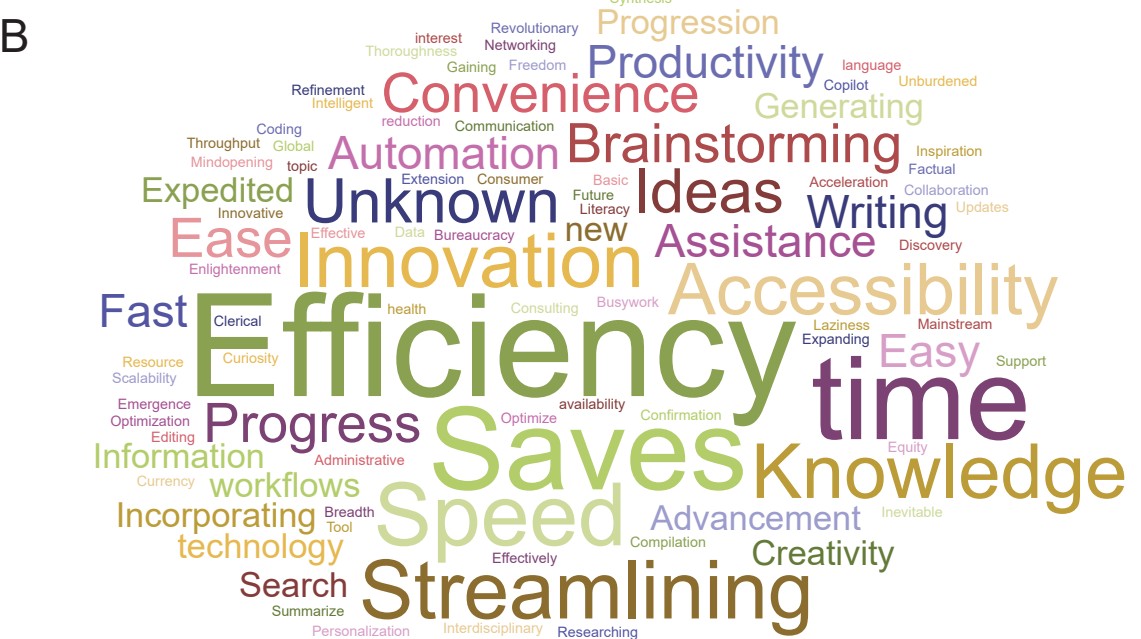

**Fig 4. Keyword responses.** (A) Respondent keywords to describe the most important risk of using ChatGPT (n = 225). (B) Respondent keywords to describe the most important benefit of using ChatGPT (n = 263).

Although medical trainees had positive views towards ChatGPT and its use, they were our smallest group of respondents (3.3% of our cohort). We took a neutral tone to the technology in preparing our recruitment material for the event, as evidenced by the respondents from other groups who had a lukewarm or uncertain view towards ChatGPT. Hence, we suspect there is high interest from medical trainees. Future studies could focus more closely on examining this group and their attitude towards LLMs.

## Conclusion

There is still much to discuss about the optimal and ethical uses of LLMs such as ChatGPT. Responsible use should be promoted, and future discussion should continue to explore the limitations of this technology. LLMs and AI in general have the potential to change the fabric of society and impact labor relations at large, deeply transforming *how we relate to one another and work*. They are like a double-edged sword, bringing with them the promise of more efficiency, creativity and free time for all, but risking spreading bias, hate, misinformation, and furthering the digital divide between those that have access to technology and are fluent in its use versus the ones left behind. The broad interest and engagement sparked by ChatGPT strongly suggests that while a work in progress, LLMs have a significant potential for disruption. To navigate this uncharted territory, we recommend that future explorations of its responsible use be grounded in principles of transparency, equity, reliability, and above all, *primum non nocere*.

## Supporting information

**S1 File. Survey delivered via Slido.** Full survey question and response options as administered to the audience.
(PDF)

## Acknowledgments

The authors wish to thank and acknowledge Eva Winckler for her contributions to event organization and coordination. We also thank the journal editor and two reviewers for their feedback.

## Author Contributions

**Conceptualization:** Mohammad Hosseini, Catherine A. Gao.

**Data curation:** Mohammad Hosseini, Catherine A. Gao.

**Formal analysis:** Catherine A. Gao.

**Investigation:** Mohammad Hosseini, Catherine A. Gao.

**Methodology:** Mohammad Hosseini, Catherine A. Gao.

**Project administration:** Mohammad Hosseini.

**Software:** Catherine A. Gao, David M. Liebovitz.

**Validation:** David M. Liebovitz, Faraz S. Ahmad, Yuan Luo, Ngan MacDonald, Kristi L. Holmes, Abel Kho.

**Visualization:** Catherine A. Gao.

**Writing – original draft:** Mohammad Hosseini, Catherine A. Gao, Alexandre M. Carvalho.

**Writing – review & editing:** Mohammad Hosseini, Catherine A. Gao, David M. Liebovitz, Alexandre M. Carvalho, Faraz S. Ahmad, Yuan Luo, Ngan MacDonald, Kristi L. Holmes, Abel Kho.

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
