## [Decision Letter · Decision Letter 0]

14 Jun 2023

PONE-D-23-13566An exploratory survey about using ChatGPT in education, healthcare, and researchPLOS ONE

Dear Dr. Hosseini,

Thank you for submitting your manuscript to PLOS ONE. After careful consideration, we feel that it has merit but does not fully meet PLOS ONE’s publication criteria as it currently stands. Therefore, we invite you to submit a revised version of the manuscript that addresses the points raised during the review process.

Please include the following items when submitting your revised manuscript:A rebuttal letter that responds to each point raised by the academic editor and reviewer(s). You should upload this letter as a separate file labeled 'Response to Reviewers'.A marked-up copy of your manuscript that highlights changes made to the original version. You should upload this as a separate file labeled 'Revised Manuscript with Track Changes'.An unmarked version of your revised paper without tracked changes. You should upload this as a separate file labeled 'Manuscript'.

We look forward to receiving your revised manuscript.

Kind regards,

Mary Diane Clark, PhD

Academic Editor

PLOS ONE

4. We note that Figure 7 in your submission contain copyrighted images. All PLOS content is published under the Creative Commons Attribution License (CC BY 4.0), which means that the manuscript, images, and Supporting Information files will be freely available online, and any third party is permitted to access, download, copy, distribute, and use these materials in any way, even commercially, with proper attribution. For more information, see our copyright guidelines: http://journals.plos.org/plosone/s/licenses-and-copyright.

a. You may seek permission from the original copyright holder of Figure 7 to publish the content specifically under the CC BY 4.0 license.

b.If you are unable to obtain permission from the original copyright holder to publish these figures under the CC BY 4.0 license or if the copyright holder’s requirements are incompatible with the CC BY 4.0 license, please either i) remove the figure or ii) supply a replacement figure that complies with the CC BY 4.0 license. Please check copyright information on all replacement figures and update the figure caption with source information. If applicable, please specify in the figure caption text when a figure is similar but not identical to the original image and is therefore for illustrative purposes only.

Additional Editor Comments:

Thank you for submitting the article to PLOS ONE, the topic is interesting and important. The reviewers provide guidance for how they feel that the manuscript needs to be edited. Please pay attention to their comments and make all of their suggested changes. I have concerns about your discussion as it feels more like a summary of your results than an interpretation of your results. As noted by San Jose State University Writing Center (http://www.sjsu.edu/writingcenter) the discussion should include three necessary steps: interpretation, analysis, and explanation. Why are your results important and where do they fit in with what we already know. The questions a discussion should address include how or why the use of Chat GPT is helpful or harmful. What is the meaning of your findings.

We look forward to your revised manuscript

Reviewers' comments:

Reviewer's Responses to Questions

**Comments to the Author**

1. Is the manuscript technically sound, and do the data support the conclusions?

Reviewer #1: Partly

Reviewer #2: Yes

2. Has the statistical analysis been performed appropriately and rigorously? 

Reviewer #1: Yes

Reviewer #2: Yes

3. Have the authors made all data underlying the findings in their manuscript fully available?

Reviewer #1: Yes

Reviewer #2: Yes

4. Is the manuscript presented in an intelligible fashion and written in standard English?

Reviewer #1: Yes

Reviewer #2: Yes

5. Review Comments to the Author

Reviewer #1: Thank you for the opportunity to review this manuscript on ChatGPT and similar AI tools. As an academic, the topic holds great relevance and has been widely discussed. However, I believe that a comprehensive review of existing literature is essential for advancing knowledge in any field. Literature reviews serve as the foundation for new research, identifying gaps, building upon previous studies, and contributing to the overall body of knowledge. Unfortunately, this manuscript suggests that the current literature is insufficient in providing a comprehensive understanding of the subject matter.

In some cases, the insufficiency of literature can be attributed to the field being relatively new or rapidly evolving. New developments and paradigms take time to be reflected in the literature, resulting in limited studies, replication, and consensus on key concepts or theories. This dynamic nature poses challenges for researchers attempting to establish a comprehensive literature base.

Recognizing the insufficiency of existing literature is crucial for addressing gaps and advancing knowledge in the field. However, this manuscript falls short in providing a full and comprehensive picture of how ChatGPT can be utilized in broader contexts, such as education, healthcare, and research. The literature on these specific applications was limited to one paragraph each, which is insufficient.

Additionally, the manuscript heavily relies on unnecessary graphics and images, which could have been condensed to reduce fluff. The demographic information could have been presented more concisely. Furthermore, the discussion section lacks a clear data interpretation process and a deeper understanding of the results. It appears that the researchers randomly selected comments without providing a comprehensive analysis.

Moreover, there were frequent APA errors throughout the manuscript, which need to be addressed.

Overall, I appreciate the opportunity to review this manuscript. However, improvements are needed to provide a more comprehensive literature review, condense unnecessary elements, and address the APA errors.

Reviewer #2: The paper is not as per journal format.

The font sizes are different in several places

Instead of bibliography it should be references

If possible Need to cite more papers related to chatGPT and related to this research

Figures titles are not as per journal standards and make sure to use any templetes related to plus one available

6. PLOS authors have the option to publish the peer review history of their article (what does this mean?). If published, this will include your full peer review and any attached files.

Reviewer #1: **Yes: **ASHLEY GREENE

Reviewer #2: **Yes: **Dinesh Kalla

---

## [Author Response · Author response to Decision Letter 0]

4 Aug 2023

Response to editor comments,

Thank you for your comments. We have revised our manuscript and included a point-by-point response to the reviewer comments below. In response to other comments raised:

Formatting: We have reformatted our document per PLOS ONE’s style requirements as requested. Our supplemental file includes a table and a list of questions both with relevant captions. Please let us know if there are additional formatting changes to be made.

Data sharing: We have published our dataset on Zenodo at doi 10.5281/zenodo.7789186 and all code needed to completely reproduce our results at Github repository https://github.com/cloverbunny/gptsurvey/blob/main/gptsurvey.ipynb. This is articulated in our Data and Code Availability statements. 

Regarding Word Cloud Figure licensing: This is not a copyrighted image, as we generated this with our specific data. We contacted the author of the website to clarify and request his signature on the provided document; he said it was not necessary for us to obtain formal documentation of use from him, see screenshot of email communication and disclaimer below. The word cloud generator website explicitly contains the permission: “The generated word clouds may be used for any purpose.” 

Response to reviewer comments:

Reviewer #1 Comment #1: Thank you for the opportunity to review this manuscript on ChatGPT and similar AI tools. As an academic, the topic holds great relevance and has been widely discussed. However, I believe that a comprehensive review of existing literature is essential for advancing knowledge in any field. Literature reviews serve as the foundation for new research, identifying gaps, building upon previous studies, and contributing to the overall body of knowledge. Unfortunately, this manuscript suggests that the current literature is insufficient in providing a comprehensive understanding of the subject matter.

In some cases, the insufficiency of literature can be attributed to the field being relatively new or rapidly evolving. New developments and paradigms take time to be reflected in the literature, resulting in limited studies, replication, and consensus on key concepts or theories. This dynamic nature poses challenges for researchers attempting to establish a comprehensive literature base. Recognizing the insufficiency of existing literature is crucial for addressing gaps and advancing knowledge in the field. However, this manuscript falls short in providing a full and comprehensive picture of how ChatGPT can be utilized in broader contexts, such as education, healthcare, and research. The literature on these specific applications was limited to one paragraph each, which is insufficient.

Reviewer #1 Response #1: We thank the reviewer for the comments and agree that this is a rapidly developing field. We have expanded upon our introduction and background in all three realms of education, healthcare, and research. We have furthermore linked to more comprehensive reviews for interested readers for detailed discussion and literature review beyond the scope of our paper. 

Reviewer #1 Comment #2: Additionally, the manuscript heavily relies on unnecessary graphics and images, which could have been condensed to reduce fluff. The demographic information could have been presented more concisely. Furthermore, the discussion section lacks a clear data interpretation process and a deeper understanding of the results. It appears that the researchers randomly selected comments without providing a comprehensive analysis.

Reviewer #1 Response #2: We have removed and condensed some of our figures and presented our demographic information in Table 1 in a concise format. We included all comments raised during the Question and Answer session without any random selection. We have expanded upon our Discussion. 

Reviewer #1 Comment #3: Moreover, there were frequent APA errors throughout the manuscript, which need to be addressed.

Reviewer #1 Response #3: PLOS One waives all formatting requirements for initial submissions, which is why we did not initially adhere strictly to these guidelines. We have made stylistic revisions in this new document and adhered to the style as outlined by PLOS. We are happy to work on any additional formatting requests from a copy editor. 

Reviewer #2 Comments: The paper is not as per journal format.

The font sizes are different in several placesInstead of bibliography it should be references If possible Need to cite more papers related to chatGPT and related to this research

Figures titles are not as per journal standards and make sure to use any templates related to plus one available

Reviewer #2 Responses: We have corrected these stylistic errors, cited more papers related to the research, and corrected figure titles and references. PLOS One waives all formatting requirements for initial submissions, which is why we did not initially adhere strictly to these guidelines. We are happy to consider additional formatting changes as suggested by a copy editor.

---

## [Decision Letter · Decision Letter 1]

4 Sep 2023

PONE-D-23-13566R1An exploratory survey about using ChatGPT in education, healthcare, and researchPLOS ONE

Dear Dr. Hosseini,

Thank you for submitting your manuscript to PLOS ONE. After careful consideration, we feel that it has merit but does not fully meet PLOS ONE’s publication criteria as it currently stands. Therefore, we invite you to submit a revised version of the manuscript that addresses the points raised during the review process.

Please include the following items when submitting your revised manuscript:A rebuttal letter that responds to each point raised by the academic editor and reviewer(s). You should upload this letter as a separate file labeled 'Response to Reviewers'.A marked-up copy of your manuscript that highlights changes made to the original version. You should upload this as a separate file labeled 'Revised Manuscript with Track Changes'.An unmarked version of your revised paper without tracked changes. You should upload this as a separate file labeled 'Manuscript'.If applicable, we recommend that you deposit your laboratory protocols in protocols.io to enhance the reproducibility of your results. Protocols.io assigns your protocol its own identifier (DOI) so that it can be cited independently in the future. For instructions see: https://journals.plos.org/plosone/s/submission-guidelines#loc-laboratory-protocols. Additionally, PLOS ONE offers an option for publishing peer-reviewed Lab Protocol articles, which describe protocols hosted on protocols.io. Read more information on sharing protocols at https://plos.org/protocols?utm_medium=editorial-email&utm_source=authorletters&utm_campaign=protocols.

We look forward to receiving your revised manuscript.

Kind regards,

Mary Diane Clark, PhD

Academic Editor

PLOS ONE

Journal Requirements:

Additional Editor Comments:

Thank you for all of the fixes in this revised manuscript. Plos One does not have copy editing so I am going to support the reviewers suggestion that you hire a copy editor to check all of the English and then work with someone who can help you edit down your table.

Her comments were

I would recommend a copy editor to do the final polishing of the manuscript, and to help create APA style tables. Currently Table 1 takes 3.5 pages and could be broken down into several smaller table to help readability.

Reviewers' comments:

Reviewer's Responses to Questions

**Comments to the Author**

1. If the authors have adequately addressed your comments raised in a previous round of review and you feel that this manuscript is now acceptable for publication, you may indicate that here to bypass the “Comments to the Author” section, enter your conflict of interest statement in the “Confidential to Editor” section, and submit your "Accept" recommendation.

Reviewer #1: All comments have been addressed

Reviewer #2: All comments have been addressed

2. Is the manuscript technically sound, and do the data support the conclusions?

Reviewer #1: Yes

Reviewer #2: Yes

3. Has the statistical analysis been performed appropriately and rigorously? 

Reviewer #1: Yes

Reviewer #2: Yes

4. Have the authors made all data underlying the findings in their manuscript fully available?

Reviewer #1: Yes

Reviewer #2: Yes

5. Is the manuscript presented in an intelligible fashion and written in standard English?

Reviewer #1: Yes

Reviewer #2: Yes

6. Review Comments to the Author

Reviewer #1: Thank you for your revisions. At this time, I feel my original comments were adequately addressed.

I would recommend a copy editor to do the final polishing of the manuscript, and to help create APA style tables. Currently Table 1 takes 3.5 pages and could be broken down into several smaller table to help readability.

Reviewer #2: All comments have being addressed and this paper can be accepted with no further changes.

All the images and tables are as per journal format.

7. PLOS authors have the option to publish the peer review history of their article (what does this mean?). If published, this will include your full peer review and any attached files.

Reviewer #1: No

Reviewer #2: **Yes: **Dinesh Kalla

---

## [Author Response · Author response to Decision Letter 1]

7 Sep 2023

Reviewer #1: Thank you for your revisions. At this time, I feel my original comments were adequately addressed. I would recommend a copy editor to do the final polishing of the manuscript, and to help create APA style tables. Currently Table 1 takes 3.5 pages and could be broken down into several smaller table to help readability.

Thanks for your suggestion. We have broken down Table 1 into two smaller tables and applied the APA style. We also had the paper reviewed and improved for readability. 

Reviewer #2: All comments have being addressed and this paper can be accepted with no further changes.

Thanks

---

## [Editor Report · Decision Letter 2]

11 Sep 2023

PONE-D-23-13566R2An exploratory survey about using ChatGPT in education, healthcare, and researchPLOS ONE

Dear Dr. Hosseini,

Thank you for submitting your manuscript to PLOS ONE. After careful consideration, we feel that it has merit but does not fully meet PLOS ONE’s publication criteria as it currently stands. Therefore, we invite you to submit a revised version of the manuscript that addresses the points raised during the review process.

Extremely minor grammar issues.  Should take about 15 minutes to get it corrected.

We look forward to receiving your revised manuscript.

Kind regards,

Mary Diane Clark, PhD

Academic Editor

PLOS ONE

Journal Requirements:

Additional Editor Comments:

Pone-d-23-13566R2

Fun paper and thanks for all of the revisions. I have a few more--must have give feedback to tooooo many doc students today. I look forward to these very minor corrections and the publication of your paper.

Abstract---Materials and Methods

Please add the word ‘a’. before Fisher’s Exact

Page 4---5th line. : embedded in used educational resources

Again page 4 first paragraph

Increased likelihood of plagiarism, propagation of irrelevant or inaccurate

information in student essays, and challenges of assessing students in the presence of

technologies such as ChatGPT are highlighted [10].

Please rephrase:

Using ChatGPT increases the likelihood of plagiarism, the propagation of irrelevant or inaccurate information in students’ essays, and challenges in assessing students’ work when technologies, such as ChatGPT, are available.

Page 5 Methods

4th line --- add ‘the’ before 27th of January 2023

Then last two lines The Northwestern IRB granted ‘an’ exemption

Notice that you are missing some articles. I have become extremely sensitive to this issue because articles are not used in ASL and my grant office gave me what for. So if this continues I will have to ask you to get a professional to check.

Page 6 8th line----platform, they are were consenting

Then under Survey Results

The smallest group was were medical trainees

Next line

Respondents), and ‘the’ second smallest by was

Page 7

Compared to 15.9% who had no interest in using it (p<0.001; Fig.1)

Page 11Possible Negative impacts

Second line

About used sources ‘used’

Page 12 last line

Such as ‘the’ Google search engine

---

## [Author Response · Author response to Decision Letter 2]

12 Sep 2023

Dear Dr. Clark,

Thank you so much for your comments and excellent suggestions. Please accept our apologies for minor formatting and grammar mistakes. 

We have reread and revised our manuscript with minor changes. We accepted all your suggestions and include a point-by-point response below.

Fun paper and thanks for all of the revisions. I have a few more--must have give feedback to tooooo many doc students today. I look forward to these very minor corrections and the publication of your paper.

Abstract---Materials and Methods

Please add the word ‘a’. before Fisher’s Exact

Done

Page 4---5th line. : embedded in used educational resources

Improved

Again page 4 first paragraph

Increased likelihood of plagiarism, propagation of irrelevant or inaccurate information in student essays, and challenges of assessing students in the presence of technologies such as ChatGPT are highlighted [10].

Improved

Please rephrase:

Using ChatGPT increases the likelihood of plagiarism, the propagation of irrelevant or inaccurate information in students’ essays, and challenges in assessing students’ work when technologies, such as ChatGPT, are available.

Improved

Page 5 Methods

4th line --- add ‘the’ before 27th of January 2023

Done

Then last two lines The Northwestern IRB granted ‘an’ exemption

Done

Notice that you are missing some articles. I have become extremely sensitive to this issue because articles are not used in ASL and my grant office gave me what for. So if this continues I will have to ask you to get a professional to check.

We have reread and improved the whole paper and paid specific attention to articles. 

Page 6 8th line----platform, they are were consenting

Done

Then under Survey Results

The smallest group was were medical trainees

Done

Next line

Respondents), and ‘the’ second smallest by was

Done

Page 7

Compared to 15.9% who had no interest in using it (p<0.001; Fig.1)

Done

Page 11Possible Negative impacts

Second line

Improved

About used sources ‘used’

Done

Page 12 last line

Such as ‘the’ Google search engine

Done

---

## [Editor Report · Decision Letter 3]

14 Sep 2023

An exploratory survey about using ChatGPT in education, healthcare, and research

PONE-D-23-13566R3

Dear Dr. Hosseini,

We’re pleased to inform you that your manuscript has been judged scientifically suitable for publication and will be formally accepted for publication once it meets all outstanding technical requirements.

Kind regards,

Mary Diane Clark, PhD

Academic Editor

PLOS ONE

Additional Editor Comments (optional):

Thank you for fixing the small issues that I noted. Congrats on a fun paper on an new and important topic.
---

## [Editor Report · Acceptance letter]

25 Sep 2023

PONE-D-23-13566R3 

An exploratory survey about using ChatGPT in education, healthcare, and research 

Dear Dr. Hosseini:

I'm pleased to inform you that your manuscript has been deemed suitable for publication in PLOS ONE. Congratulations! Your manuscript is now with our production department. 

Kind regards, 

on behalf of

Dr. Mary Diane Clark 

Academic Editor

PLOS ONE